# Evaluation of factors associated with HIV self-testing Acceptability and Uptake among the MSM community in Nairobi, Kenya: A cross sectional study

Kingori Ndungu[1]*, Peter Gichangi[1,2,3], Marleen Temmerman[1,3,4]

1 Faculty of Medicine and Health Sciences, Department of Public Health and Primary Care, Ghent University, Ghent, Belgium, 2 University Administration, Technical University of Mombasa, Mombasa, Kenya, 3 International Centre for Reproductive Health, Mombasa, Kenya, 4 Department of Obstetrics and Gynaecology, Aga Khan University Hospital, Nairobi, Kenya

* kingoriindungu@gmail.com

## Abstract

### Background

Human Immunodeficiency Virus self-test (HIVST) refers to a process where a person collects his or her own specimen (blood or oral), performs a test and interprets the results. The interpretation of results can either be done in private or through support of a trusted partner. Self-test should be seen as screening and confirmatory tests are typically strongly encouraged.

### Study objective

To determine facilitating factors for HIVST acceptability and uptake among men who have sex with men (MSM).

### Methods

A cross-sectional exploratory study design, targeting MSM in Nairobi was used. Adult men (aged 18–60 years) who reported to be actively engaging in anal or oral sex with men were eligible for the study. Purposive sampling was used to identify the sites where data was collected, snowballing technique was then employed to reach the respondents. Data was collected between July 2018 and June 2019. A total of 391 MSM respondent were recruited of whom 345 MSM completed the questionnaires. The missing data was handled through the listwise approach that omits those cases with the missing data and analyze the remaining data. We also excluded responses with inconsistencies in all confirmatory questions in the questionnaire.

### Results

Two-thirds (64.0%) of the participants were aged 18–24 years with 13.4% being married to women and 40.2% having tertiary level of education. Majority, 72.7% were unemployed and

**Data Availability Statement:** All relevant data are within the paper and its Supporting Information files.

**Funding:** The author(s) received no specific funding for this work.

**Competing interests:** The authors have declared that no competing interests exist.

**Abbreviations:** ART, Antiretroviral Therapy; AIDS, Acquired Immunodeficiency Syndrome; HIV, Human Immunodeficiency Virus; HIVST, Human Immunodeficiency Virus self -test; HTS HIV, Testing Services; LMIC, Low and middle- income countries; MSM, Men who have sex with men; MSW, Men who are sex workers.

two-thirds (64.0%) of participants were young (18–24 years) and self reported as male sex workers (58.8). There were significant associations between willingness to undertake HIV self-tests and frequency of HIV testing as well as with previous knowledge about self-testing. Habitual HIV testers were more likely to have used the HIVST kit than the non-habitual testers. Willingness to undertake confirmatory test within one month of self-testing was associated with acceptability of HIVST. Most of the MSM preferred blood sample self-test kits as compared to oral self-test kits, believing that blood test will be more accurate than oral self-test. Other factors associated with HIVST included consistent use of protection regardless of HIV status, preference of "treatment buddies". High costs of the self-test kits and inadequate knowledge on the use of HIV self-test kits were the main hindrances to HIVST uptake.

## Conclusions

This study has showed that age, habitual testing, self-care/partner care, as well as confirmatory testing and immediate introduction into care if found sero-positive were associated with the use of HIVST kit. This study contributes to the pool of knowledge of the characteristics of MSM that would adopt and embrace HIVST, and demonstrates that these MSM are self and partner care aware and conscious. The challenge however remains on how to encourage those that are not self/partner care aware to embrace HIV testing and particularly HIVST as routine practices. Future studies may need to explore potential motivators to self-testing among the young, elder MSM generations and the MSM with higher economic status in Kenya.

## Background

There has been significant progress in HIV prevention efforts across Africa [1–5], however men who have sex with men (MSM) continue to bear a disproportionately heavy burden of HIV infection compared to the general adult population [6]. The risk of acquiring HIV is 27 times higher among MSM compared to other heterosexual men [7]. Surveillance data from low and middle-income countries (LMIC), shows that MSM are 19.3 times more likely to be living with HIV than the general population [8]. Kenya has the third largest population of people living with HIV in sub-Saharan Africa and the highest national HIV prevalence of any country outside of Southern Africa [9]. HIV in Kenya is characterized as a generalized epidemic among the adult population, and a concentrated epidemic among key populations [10]. The MSM population in Kenya commonly encompasses a range of sexual and gender identities, including homosexual, gay, bisexual, transgendered, and heterosexual. MSM may be married, gay, or sex workers [11]. The Kenya AIDS Strategic Framework 2014/15–2018/19 identified MSM and male sex workers (MSWs), as key populations and has prioritized these populations for HIV/AIDS prevention [12]. HIV prevalence among the MSW and MSM in Kenya is 4 to 5 times higher than in the among the general population [11]. HIV prevalence in 2010 among MSM in Kenya was an estimated 18.2% [13]. According to the Kenya HIV Prevention Response and Modes of Transmission Analysis (2008), 15.2 percent of new infections were attributed to MSM [14]. In 2007, HIV prevalence among MSM/MSWs with exclusively male partners was 41 percent in Mombasa [15]. The above findings further confirm the

assertion by the Kenya AIDS Strategic Framework 2014/15–2018/19, that the MSM and MSW should be prioritized in controlling the HIV epidemic in Kenya.

More than half (53%) of the 1.6 million people living with HIV in Kenya are unaware of their HIV status [16]. The 2019 HPTN 075 study, aimed at evaluating the accuracy of self-reported HIV status among African men who have sex with men, indicated that self-report of HIV positive status at screening was less common among participants who reported sex with men and women compared to men who reported sex with men only, perhaps due to concerns about inadvertent disclosure about their same-sex behaviors [17]. In this report, approximately one third of the HIV-infected MSM who did not report a prior HIV diagnosis to study staff had ARV drugs detected and were classified as previously diagnosed [18]. It is estimated that new HIV infections could be reduced by 30% per year, if all HIV positive individuals were aware of their serological status [19].

The World Health Organization (WHO) recommends that all MSM test for HIV at least annually and those MSM who have multiple or anonymous partners or use illegal drugs should test at least every 3 or 6 months [20]. Additionally, the United Nations Programme on HIV/ AIDS (UNAIDS) advocates that at least 90% of people should know their HIV status, 90% of those diagnosed linked to anti-retroviral (ARV) treatment and 90% of those on treatment achieving viral suppression [21]. Achieving "the first 90%" is a critical step in the elimination of new infections by 2030 [22]. However, more than 75% of MSM in low and middle-income countries lack access to HIV testing services [23]. There is therefore urgent need to develop and implement innovative initiatives that promote HIV testing, especially among the MSM community. HIV self-testing (HIVST) is a process where an individual conducts a HIV test on their own specimen (oral fluid or blood), and interprets the result. HIVST can be performed in a private setting, either alone or with a trusted person. Since HIV self-testing does not provide a definitive diagnosis, individuals whose tests positive for HIV are advised to seek confirmatory testing with a health provider, in line with the national HIV testing algorithm [24].

Recent research conducted in Zimbabwe, Zambia, Lesotho and Kenya revealed that HIVST is feasible, acceptable among health workers and the general population [18,22,23,25–27]. The Kenyan government introduced self-testing kits as part of "Be Self Sure campaign in May, 2017. The kits are now available to buy from pharmacies for around 8–10 United States Dollars (USD). There is potential for HIVST to enhance access to HIV testing especially within the MSM community [28], even though uptake of HIV Testing Services (HTS) among the MSM is low, compared to those in the general population due to barriers such as stigma, discrimination and poor quality of services [29].

Countries around the world are at different stages of development and implementation of HIVST guidelines. There is however no optimal approach for implementing HIVST targeting the MSM community. Very few studies have examined any aspect of HIVST among MSM in Sub-Saharan Africa [26,27]. In Kenya studies have been conducted to understand the feasibility of scaling up HIVST within the general population, including among sero-discordant couples and FSWs, yet there is little information on HIVST among MSM specifically [28,29]. Although global evidence suggests that introduction of HIVST can benefit MSM, evidence is limited on how, when, and in what contexts the delivery of HIVST to MSM could increase awareness of HIV status, and lead to early linkage to HIV treatment and prevention services in the African context [30]. Innovative interventions geared towards promoting testing among the MSM community must overcome key barriers to testing such as stigma, fear about confidentiality of venue testing, distance to testing sites and opportunity cost [31]. We conducted this research to explore acceptability and factors that would facilitate uptake of HIV self- testing among the MSM community in Nairobi.

## Methods

### Study design & population

The Conceptual Framework combining AIDS Risk Reduction Model (ARRM) and Modified Social Ecological Model showing the interconnectedness of the different factors informed this exploratory study [32], S1 Fig in S1 Graph. A cross-sectional design in peri-urban settings of Nairobi and Kiambu counties in Kenya was used. The study target population was all adult men above 18 years living in Nairobi county, who self-reported having engaged in anal or oral sex with men, and were able to provide own informed consent. Nairobi city has the highest number of MSM/MSW, approximately 11,042 MSM (range 10,000–22,222) [33], Kiambu borders Nairobi, hence a likelihood of interviewing residents from Kiambu in Nairobi county.

### Sampling techniques

Purposive sampling to identify the eleven data collection points frequented by the MSM community, such as the safe spaces/drop-in centres, hotels and massage parlors was used. Based on previous work in Kenya and elsewhere that indicates high mobility of this population, and given that hidden populations have networks within which they operate, data collections points which were likely to capture the same respondents or respondents with similar characteristics were not included in the study [34–40]. Snow balling or chain-referral sampling was then used to reach the interviewees [36,37,39]. This method, despite its inherent flaws, was the most optimal to use to identify and reach this hidden population. The method has been previously used to reach similar populations elsewhere [41–43]. The study reached 391 MSM who responded to the self-administered questionnaires out of which 345 completed the information. The missing data was handled through the listwise approach that omit those cases with the missing data and analyze the remaining data. This approach is known as the complete case or listwise deletion. Listwise is a default option for analysis in most statistical software packages including the Statistical Package for the Social sciences (SPSS), we used for the analysis. However, if the assumption of missing is completely at random (MCAR) is satisfied, a listwise deletion is known to produce unbiased estimates and conservative results [44]. The missing data for this study was random, hence produced unbiased estimates as demonstrated in the probability plot of normality tests graphs.

### Data collection

The questionnaire was developed through a consultative, participatory approach by consulting experts in the subject matter and the MSM/MSW. The questionnaire development process was also informed by previous research done on the thematic area [45–47]. The draft questionnaire was tested with a sample of 45 respondents for flow, comprehension, conciseness and the ability to adequately answer the research questions. All the MSM respondents completed the English paper based self-administered structured questionnaires between July 2018 and June 2019 after obtaining oral informed consent. These structured questionnaires consisted of three sections. The first section had questions on demographics of the respondents, the second section consisted of questions on HIV risk behaviors and the final section had question on HIVST preferences, acceptability and uptake.

### Measures

The dependent variable for the study was reported HIV self-testing. Self-reports on HIVST have been used in previous studies, either with or without verification that the tests were actually conducted as reported [48–56]. Prior to data collection, the meaning of HIV self-test was

explained to the participants and it included a test that allows people to take a HIV test and find out their result in their own home or other private locations. Participants were then requested to indicate whether they had ever undertaken a HIV self-test prior to the study.

The independent variables included socio-demographics: Age, place/country of birth, religion, education level, employment status, marital status and income. While it is possible that there were false reports, checks were put in place to reduce the possibility of wrong reporting by excluding responses with inconsistencies in all the cascades. The questionnaires were also pre-tested and refined before the main data collection.

Besides the above, the following risk factors data were collected: Identify as MSM/MSW, number of sexual partners, preferred sex position (top or bottom), condom use, condom use after alcohol/hard drugs and type of lubricant. The following variables were of importance in measuring the acceptability and factors that facilitates the uptake of HIVST: Frequency of HIV/AIDS test, use of protection or condoms with partner if HIV/AIDS negative, ever heard of HIV self-test, would prefer to use the Oral or Blood self-test kit, cost of the HIV self-test kits, ever tested for HIV, type of facility for the last HIV test, reason for most recent test, ever heard of window period, and have you ever experienced stigma from healthcare providers.

## Data analysis & management

The questionnaires were serialized and data entered in a SPSS Version 23.0 data-base for analysis (IBM Corp, 2015). Using descriptive analysis, we summarized and presented data in tables. Cross tabulation and bivariate analysis were done. Prevalence Odds Ratio (POR) and multivariate analysis was done to get adjusted POR including variables statistically significant on bivariate analysis or those known to have impact on the variable of interest.

P-value of <0.05 was considered statistically significant.

## Ethical considerations

Ethical clearance was obtained from the University of Ghent Approval number (PA 2016/009) and the Mount Kenya University Ethics Review Committee (Approval number: MKU/ERC/ 0463). Informed consent was obtained from all study participants in a language that they could understand. We anonymized each record using unique identifiers during data entry and analysis. Access to the data was restricted to only those researchers responsible for analysis in password protected databases and computers.

## Results

### Demographics, social-economic factors

About (86.6%) of the interviewees were Kenyans. Two-thirds (64.0%) of the interviewees were aged 18–24 years of whom 86.6% were single and only 13.4% were married. A majority of the interviewees were Christians 244 (71.8%). Education attainment was high with (59.8%) reporting at least secondary education and (40.2%) tertiary level of education, however, employment status was low with (72.7%) reporting that they were unemployed. Slightly more than half (52.8%) reported earning more than 60 United States Dollars (USD), while (47.2%) earned less than 60 USD per month. More than half of all respondents (58.8%) self-identified as MSW, while the rest (41.2%) identified as MSM, (Table 1).

### Uptake of HIV self testing

Of all the participants, (55.9%) reported HIV self-testing. More older respondents aged >25 years (53.51%) reported having ever used HIV test kit, than those aged between 18–24 years

**Table 1. Respondent demographic & social economic information.**

| Variable | | Frequency | Percent |
|---|---|---|---|
| Respondent's age in years | 18–24 | 203 | 64.0 |
| | 25 + | 114 | 36.0 |
| Place of Birth | Kenya | 291 | 86.6 |
| | Non-Kenyans | 45 | 13.4 |
| Religion | Christian | 244 | 71.8 |
| | Non- Christian | 96 | 28.2 |
| Education level | At least Secondary education | 204 | 59.8 |
| | Tertiary | 137 | 40.2 |
| If currently employed | Employed | 47 | 27.3 |
| | Unemployed | 125 | 72.7 |
| Monthly income | Less 6,000 | 94 | 47.2 |
| | 6,000 + | 105 | 52.8 |
| Identity as MSM or MSW | MSM | 142 | 41.2 |
| | MSW | 203 | 58.8 |
| Marital status | Single | 291 | 86.6 |
| | Married | 45 | 13.4 |

MSM-men who have sex with men MSW-men who are sex workers.

(39.9%), POR 1.73 (1.09, 2.75). Some of the HIV risk behaviors for the interviewees include having multiple sexual partners and being MSW. A higher proportion of the interviewees (90.8%) affirmed that they used condoms during the last sex act. However, (57.7%) consistently used condoms. Regarding the preference in sex position, most of the interviewees were versatile, they preferred being either on top or bottom. Comparatively fewer Kenyans (42.96%) than non-Kenyans (57.78%) indicated that they had ever used HIV self-test kit. Cross-tabulation analysis of ever used HIV test kit by employment status showed that (40.43%) of the employed and (47.2%) of the unemployed had used the kit POR 1.32 (1.67, 2.60). There was a higher chance of employed respondents using the HIV test kits at (32%.) than the unemployed. The interviewees who were single were (86.61%) and married (13.39%) had statistical significance with POR 1.82 (1.96, 3.43) one use of HIV self test kit. There were more study participants who reported having used a condom during last sex and had used HIV test kit (47.21%), than those that had not used a condom during sex and had ever used an HIV test kit (29.03%) POR 2.19 (1.98, 4.90). Participants who reported condom use at last sex were nearly 2-times more likely to have had a self- HIV testing POR 2.19 (1.98, 4.90). Half the respondents (50.63%) indicated that they considered the window period in deciding when to test POR 1.47 (1.96, 2.25). The participants who reported that if they test positive on self-HIV testing were more than 2-times likely to report they would use protection or condoms with their partner POR 1.32 (1.67, 2.6). On univariate analysis, sex position, POR 1.01 (0.94, 1.08) did not influence uptake of HIV self-testing, Table 2.

## HIV self-test acceptability, uptake & associated factors

About (77%) of the respondents were willing to self test for HIV. Among the factors influencing HIV self-testing that had a statistically significant association at Pearson Chi-Square P-value <0.05 at 95% confidence interval were frequency of HIV/AIDS test, considered the window period in deciding when to test, use protection or condoms with partner if HIV/AIDS positive, ever heard of HIV self-test, ever heard of oral HIV Self-test, heard of blood sample

**Table 2. Uptake of HIV self -testing & risk factors.**

| Variable | | Ever Self -Tested | | Total | Pearson Chi-Square | Prevalence Odds Ratio (95% Confidence Interval) |
|---|---|---|---|---|---|---|
| | | No | Yes | | | |
| Respondent's age in years | 18–24 | 122 (60.1) | 81 (39.9) | 203 (64.04) | 0.020 | Ref |
| | 25 + | 53 (46.49) | 61 (53.51) | 114 (35.96) | | 1.73 (1.09,2.75) |
| Place of Birth | Kenya | 166 (57.04) | 125 (42.96) | 291 (86.61) | 0.045 | Ref |
| | None Kenyan | 19 (42.22) | 26 (57.78) | 45 (13.39) | | 1.82 (1.6,3.43) |
| Education level | At least Secondary | 105 (51.47) | 99 (48.53) | 204 (59.82) | 0.126 | Ref |
| | Tertiary | 80 (58.39) | 57 (41.61) | 137 (40.18) | | 0.76 (0.49,1.17) |
| Monthly income | Less 60 USD | 43 (45.74) | 51 (54.26) | 94 (47.24) | 0.144 | Ref |
| | 60 USD + | 57 (54.29) | 48 (45.71) | 105 (52.76) | | 0.71 (0.41,1.24) |
| Current employment status | Employed | 28 (59.57) | 19 (40.43) | 47 (27.33) | 0.014 | Ref |
| | Unemployed | 66 (52.8) | 59 (47.2) | 125 (72.67) | | 1.32 (1.17,2.60) |
| Marital status | Single | 166 (57.04) | 125 (42.96) | 291 (86.61) | 0.045 | Ref |
| | Married | 19 (42.22) | 26 (57.78) | 45 (13.39) | | 1.82 (1.69,3.43) |
| Sexual partners last 6 months | One | 36 (52.17) | 33 (47.83) | 69 (20.66) | 0.426 | Ref |
| | More than one | 144 (54.34) | 121 (45.66) | 265 (79.34) | | 0.92 (0.54,1.56) |
| Prefer Top or Bottom | Top | 6 (5.7) | 98 (94.2) | 104 (89.6) | 0.068 | Ref |
| | Bottom | 6 (10.3) | 52 (89.6) | 58 (50) | | 1.01 (0.94,1.08) |
| Use condom during last sex | Yes | 161 (52.79) | 144 (47.21) | 305 (90.77) | 0.039 | 2.19 (1.98,4.90) |
| | No | 22 (70.97) | 9 (29.03) | 31 (9.23) | | Ref |
| If yes how often do you use a condom during sex | Consistently | 90 (50.28) | 89 (49.72) | 179 (57.74) | 0.167 | Ref |
| | Inconsistently | 74 (56.49) | 57 (43.51) | 131 (42.26) | | 0.78 (0.50,1.23) |
| Type of lubricant in last sexual act | KY Jelly | 94 (49.47) | 96 (50.53) | 190 (64.85) | 0.094 | Ref |
| | None water-based | 60 (58.25) | 43 (41.75) | 103 (35.15) | | 0.7 (0.43,1.14) |
| Do you use a condom during anal sex after alcohol/ hard drug use | Consistently | 71 (46.1) | 83 (53.9) | 154 (52.74) | 0.099 | Ref |
| | Inconsistently | 75 (54.35) | 63 (45.65) | 138 (47.26) | | 0.72 (0.45,1.14) |
| Have you ever had any mental health issue | Yes | 14 (41.18) | 20 (58.82) | 34 (10.33) | 0.073 | 1.81 (0.88,3.73) |
| | No | 165 (55.93) | 130 (44.07) | 295 (89.67) | | Ref |

*(Continued)*

**Table 2.** (Continued)

| Variable | | Ever Self-Tested | | Total | Pearson Chi-Square | Prevalence Odds Ratio (95% Confidence Interval) |
|---|---|---|---|---|---|---|
| | | No | Yes | | | |
| Considered the window period in deciding when to test | Yes | 79 (49.38) | 81 (50.63) | 160 (46.38) | 0.048 | 1.47 (1.19,2.25) |
| | No | 109 (58.92) | 76 (41.08) | 185 (53.62) | | Ref |
| If HIV positive through HIV self-test, would you go for a confirmatory test | Yes | 150 (53.96) | 128 (46.04) | 278 (80.58) | 0.394 | 1.12 (0.65,1.91) |
| | No | 38 (56.72) | 29 (43.28) | 67 (19.42) | | Ref |
| If you test positive, would you use protection or condoms with your partner | Yes | 158 (52.32) | 144 (47.68) | 302 (87.54) | 0.022 | 2.1 (1.06,4.19) |
| | No | 30 (69.77) | 13 (30.23) | 43 (12.46) | | Ref |

HIV- Human Immunodeficiency Virus USD-United States Dollar.

HIV Self-test, would prefer to use the Oral or Blood self-test kit and what do you think of the current cost, Table 3.

## Multivariate analysis for HIV self-testing among MSM

We used the measure of association (Chi-Square) and Prevalence Odds Ratio (POR) to inform the variables included in the multivariate analysis. The covariates of HIV self-testing were: respondent's age, MSM aged 25 and above were more likely to have used HIVST than those 24 years and below; place of birth, frequency of testing for HIV/AIDS- habitual testers were more likely to have used HIVST kits than non-habitual testers; if HIV positive through HIV self-test, would go for a confirmatory test, if confirmatory test shows negative, after how long would you seek a re-test after the current negative result test. Those that would test regularly even if previous HIV test was negative, were more likely to have used the test kit, Table 4.

## Discussion

The findings of this study significantly contribute towards the generation of evidence on HIV self-testing among the MSM in Kenya as proposed by World Health Organization (WHO) 2016 guidelines [57]. This study majored on HIVST among MSM and MSW in Nairobi and the peri-urban neighboring.

The study suggests that there is a significant association between HIV self-testing and frequency of testing. MSM who used HIV self-test kits had a higher frequency of testing. Habitual HIV testers were more likely to have used the HIVST kit than the non-habitual testers and this finding is consistent with other findings that suggest that regular testing is often associated with other protective measures such as education, fewer sexual partners as well as the mindset to care for self and partners [58,59]. Our findings are also in agreement with MSM randomized to HIVST access Vs. standard clinic-based testing in Seattle, the mean number of HIV test and quarterly testing increased significantly among those in the HIVST, with no increase in risk behaviors [60]. Kenya's National HIV testing guidelines recommends re-testing of HIV negative MSM every three months [61]. To achieve the above target, both clinic-based setting and HIV self-test options should be scaled up within the MSM community.

Our study findings show that most of the MSM preferred blood sample self-test kits as compared to oral self-test kits, most of the interviewees preferred blood sample self-test kits, since

**Table 3. HIV self-testing acceptability, uptake & associated factors.**

| Variable | | Used HIV test kit before | | Total | Pearson Chi-Square | Prevalence Odds Ratio (95% Confidence Interval) |
|---|---|---|---|---|---|---|
| | | No | Yes | | | |
| Ever tested for HIV/AIDs | No | 11 (64.71) | 6 (35.29) | 17 (4.93) | Ref | 0.76 (0.40,1.48) |
| | Yes | 177 (53.96) | 151 (46.04) | 328 (95.07) | 0.386 | |
| If yes how often do you test for HIV/AIDS | Every 3 months | 118 (50.43) | 116 (49.57) | 234 (67.83) | Ref | 1.34 (1.02, 1.77) |
| | Greater than 3 months | 70 (63.06) | 41 (36.94) | 111 (32.17) | 0.028 | |
| Results of the most recent HIV test | Negative | 138 (52.67) | 124 (47.33) | 262 (75.94) | Ref | 1.19 (0.89,1.60) |
| | Positive | 50 (60.24) | 33 (39.76) | 83 (24.06) | 0.228 | |
| Ever heard of window period | No | 102 (57.63) | 75 (42.37) | 177 (51.30) | Ref | 0.87 (0.69,1.09) |
| | Yes | 86 (51.19) | 82 (48.81) | 168 (48.70) | 0.360 | |
| If yes after how long would you go for a confirmatory test | Won't go | 14 (51.85) | 13 (48.15) | 27 (7.99) | Ref | 0.93 (0.62,1.40) |
| | Within a Month | 172 (55.31) | 139 (44.69) | 311 (92.01) | 0.729 | |
| Consider taking up HIV self test as one of the HIV combination prevention strategy | No | 47 (59.49) | 32 (40.51) | 79 (22.90) | Ref | 0.86 (0.64,1.16) |
| | Yes | 141 (53.01) | 125 (46.99) | 266 (77.10) | 0.309 | |
| Would go for counselling after positive result | No | 105 (58.01) | 76 (41.99) | 181 (52.46) | Ref | 0.85 (0.68,1.07) |
| | Yes | 83 (50.61) | 81 (49.39) | 164 (47.54) | 0.168 | |
| Ever heard of HIV self test | No | 82 (65.08) | 44 (34.92) | 126 (36.52) | Ref | 0.68 (0.52, 0.89) |
| | Yes | 106 (48.40) | 113 (51.60) | 219 (63.48) | 0.003 | |
| Ever heard of oral HIV Self test | No | 101 (66.45) | 51 (33.55) | 152 (44.06) | Ref | 0.61(0.47, 0.79) |
| | Yes | 87 (45.08) | 106 (54.92) | 193 (55.94) | 0.001 | |
| Heard of blood sample HIV Self test | No | 78 (77.23) | 23 (22.77) | 101 (29.28) | Ref | 0.42 (0.29, 0.60) |
| | Yes | 110 (45.08) | 134 (54.92) | 244 (70.72) | 0.001 | |
| Would prefer to use the Oral or Blood self test kit | Oral self test kit | 76 (61.79) | 47 (38.21) | 123 (42.27) | Ref | 0.76 (0.58, 0.99) |
| | Blood self test kit | 83 (49.40) | 85 (50.60) | 168 (57.73) | 0.036 | |
| If using self test for the first time would you prefer to have a treatment "buddy" | No | 26 (54.17) | 22 (45.83) | 48 (15.43) | Ref | 1.01 (0.72,140) |
| | Yes | 143 (54.37) | 120 (45.63) | 263 (84.57) | 0.490 | |
| What do you think of the current cost | Affordable | 101 (69.18) | 45 (30.82) | 146 (42.32) | Ref | 0.55 (0.42, 0.72) |
| | Expensive | 87 (43.72) | 112 (56.28) | 199 (57.68) | 0.001 | |

*(Continued)*

**Table 3.** (Continued)

| Variable | | Used HIV test kit before | | Total | Pearson Chi-Square | Prevalence Odds Ratio (95% Confidence Interval) |
|---|---|---|---|---|---|---|
| | | No | Yes | | | |
| Preference point of picking the oral HIV self test kit or what distribution channel would you prefer | NGO | 130 (54.90) | 107 (45.10) | 237 (77.50) | Ref | 0.94 (0.71,1.25) |
| | Government | 36 (52.20) | 33 (47.80) | 69 (22.50) | 0.398 | |

HIV-Human Immunodeficiency Virus AIDS- Acquired Immunodeficiency Syndrome.

they believe the blood test will be more accurate than oral self-test. These findings are in line with a study conducted in South Africa among the MSM community that showed higher preference for fingerstick tests over oral fluid tests among the interviewees [62]. Our findings differ from study findings conducted in the US emergency department, where most of the respondents prefered oral fluid testing [63]. Given the diversity of preference for either blood sample or oral test kits and the ovewhelming support for HIVST, we would highly recommend the availability and distibution of both types of self kits to the MSM community in Kenya.

The Non-Govermental Organisations (NGO) facilities/drop in centres were the most preferred distibution points for the HIV self- test kits. Other distibution points include the private sector and the public/government facilities. Our findings differ slightly with a study conducted by Okal et al., [64], that's showed that most of the respondents preferred public health care facilities but for the general population. The variance in the findings can be attributed to the fact that the MSM community feel stigmatized and also due to the illegal nature of homosexuality in Kenya, hence the MSM would want to stay far away from the public/Government healthcare providers. Site preference is also largely based on proximity and cost. A study conducted among general adult population showed that "easily available" as the strongest reason

**Table 4. Multivariate analysis of covariates for HIV self-testing among MSM.**

| Variables in the equation | P-Value | APOR | Lower | 95% C.I for APOR |
|---|---|---|---|---|
| | | | | Variables in the equation |
| Respondent's age | 0.048 | 2.986 | 2.75 | 3.81 |
| Place of Birth | 0.044 | 1.181 | 1.11 | 1.29 |
| Religion | 0.173 | 1.228 | 0.34 | 3.73 |
| Education level | 0.285 | 2.904 | 0.49 | 3.73 |
| Current employment status | 0.212 | 0.219 | 0.46 | 3.32 |
| Monthly income | 0.635 | 1.521 | 0.15 | 1.75 |
| Sexual partners last 6 months | 0.555 | 1.124 | 0.10 | 1.28 |
| Do you use condoms during anal sex after alcohol/hard drug use | 0.133 | 3.454 | 0.60 | 5.06 |
| How often do you test for HIV/AIDS | 0.003 | 1.243 | 1.11 | 1.34 |
| Ever heard of the window period | 0.192 | 3.783 | 0.48 | 3.81 |
| Consider taking up HIV self-test as one of the HIV combination prevention strategies | 0.696 | 0.645 | 0.30 | 3.23 |
| If HIV positive through HIV self-test, would you go for a confirmatory test | 0.042 | 1.680 | 1.04 | 1.89 |
| If yes after how long would you go for a confirmatory test | 0.729 | 1.602 | 0.13 | 5.44 |
| If the confirmatory test shows negative, after how long would you seek a re-test after the current negative result test | 0.049 | 1.670 | 1.16 | 2.58 |

APOR-Adjusted prevalence odds ratio C.I-Confidence Interval.

for which ever pick up point [65]. A study conducted in Nigeria found that MSM prefer to pick the kits from the MSM friendly drop in centers, and community health clinic (CHC) [48]. Future HIVST distribution should consider drop in centers and community pharmacies as options of distribution sites. Peer educators or Key Opinion Leaders (KOL) are still an option to be considered, though this approach would have high cost implications.

A high proportion of the respondents (80%) indicated that if they tested for HIV positive through HIV self-test, they would go for counselling and confirmatory test. Our findings are consistent with findings from a cross-sectional study done in Kenya, that showed that 74% of the respondents would seek counselling services, confirm results or seek medication after a positive HIVST [66]. Pre-test counselling should be offered before dispensing the HIV self-test. This would provide an opportunity for client to get all the information they would need before testing. The provision of information before testing would be crucial in guiding the clients on how to use the test kits, so as to reduce invalid results also the clients would know what to do depending on the results and where to seek for the healthcare services. Pre-test counseling data can also be used for monitoring the patients and for follow-ups.

A significantly high proportion of the respondents (92%) would go for confirmatory test within one month of testing. Our findings are slightly higher than a study conducted in Kenya in 2014, [66] that reported 61% of the general population sampled and 40% of MSM would go to a clinic for a confirmatory test. This demonstrates that the MSM who self-test are willing and will still link to the healthcare system. A major concern for HIVST is whether self-testers will seek care and treatment depending on the results. Being able to link to care within a week is considered optimal behavior given that linkage to care is defined as "having visited a health care provider within 30 days of being diagnosed with HIV [67]. There is therefore a challenge and a potential in enhancing uptake of HIVST among MSM. The opportunity is that, during the implementation of HIV self-test, home visits by the healthcare workers, encouraging the MSM to visit MSM friendly clinics, and calls and short message service (SMS) can potentially be utilized to ensure that the MSM that test HIV positive are not only linked to care, but are not lost in follow-up. A study conducted in Nigeria among the MSM community showed a 100% linkage to HIV and treatment [68], indicates potential viability of the suggested strategy to link and retain MSM in care. The high linkage in that study was likely due to follow-up calls after HIVST distribution, the ease of participant's access to the opinion leaders and also the linkage to a well-trusted MSM-friendly facility that offers HIV prevention services. Studies conducted in other areas reported both fairly low [69] and high [70] linkage to HIV care and treatment after self-testing.

The use of a "test buddy" may potentially increase uptake of HIVST. A significantly high number of MSM indicated that they would prefer a testing "buddy" on the first time of self-test kit use. There was a statistically significant association between respondents who had never used a HIV self-test kit and the respondents' need of a testing partner on first time use of HIV self-test kit. A HIVST validation study was conducted in Kenya, that reported a higher rate of invalid HIV results (37/ 239 = 15.5%) [65]. The findings are also consistent with another study done among Chinese MSM, that showed significant errors during the process of conducting HIVST that rendered almost half of the test results invalid [71]. According to that study, failure to follow the manufacturers instructions was the main cause of invalid results both for oral self test and blood sample test kits. For the finger prick users, most of the errors occurred during the stage of collecting specimen and for oral fluid users made most of the errors during the stage of testing the collected specimen. A testing partner or "buddy" would also help the MSM first time testers get accurate results.

The high cost of the test kits, lack of knowledge on correct usage of the test kits, fear, stigma and inaccessibility of HIV self-test kits were the main hindrances on the use of the kits. Most

of the respondents indicated that the estimated current market cost of the self-test kit was expensive at 8–100 USD. The preferred cost of the HIVST kit was about half to one USD. Lack of appropriate knowledge on the use of HIVST kits would prevent the MSM from using the kits. This correlates to studies showing several ways in which errors would happen, this would include during sample collection using the swab, also the handling of the swab and following the procedures [65,72].

This study has some limitations. The study is based on respondent self-reports on HIVST use, and as such the veracity of such reports could not be easily determined within the design of this study. Despite the fact that the respondents, were recruited through methods designed to generate a representative sample, the sample is unlikely representative of all MSM in Kenya, since most of the respondents were young and of low and medium economic status. Demonstrating that the older and MSM with higher economic status were not represented. There is likelihood of bias in the sample as this was response driven and typically referrals through such a strategy implies that potential respondents will be drawn from friends, colleagues and known networks of the individuals making such referrals. The respondents were also drawn from Nairobi and the neighboring county, where the MSM community has been receiving considerable support from the NGOs as compared to MSM in other regions of the country. Future studies should prioritize the elder generations and also the MSM with higher income status and also from other regions of the country with few HIV prevention interventions.

## Conclusions

This study has showed that age, habitual testing, self-care/partner care, as well as confirmatory testing and immediate introduction into care if found seropositive were associated with having used an HIVST kit. All the above are factors that potentially would also encourage prompt enrollment into linkage to care and treatment after self-test. Some of the key facilitating factors for HIVST uptake among the MSM community in Kenya include; awareness creation on the importance of frequent HIV testing, counselling pre and/ or post testing, emphasizing the importance of testing buddies for first- time testers and reducing the cost of the current HIV self-test kits. A potential strategy to increase the uptake of HIV-self testing would be using sexual and social networks as an entry point. This is based on the conceptual framework, the importance of testing buddies as well as the existence of sexual networks through which respondents were identified and recruited. Blood sample test kits are more preferred than Oral test kits. This study contributes to the pool of knowledge on the characteristics of MSM that would adopt and embrace HIVST, and demonstrates that the MSM are self and partner care aware and conscious. The challenge however remains on how to encourage those that are not self/partner care aware to embrace HIV testing and particularly HIVST as routine practices. Future studies may need to explore potential motivators to self-test among the young, elder MSM generations and the MSM with higher economic status.

## Supporting information

**S1 Study tool. Study questionnaire.**
(DOCX)

**S1 Data. Study dataset.**
(XLSX)

**S1 Graph. Theoretical framework.**
(PDF)

**S1 Plot. Probability plot.**
(PDF)

## Acknowledgments

We thank all participants for supporting this study.

## Author Contributions

**Conceptualization:** Kingori Ndungu, Peter Gichangi, Marleen Temmerman.

**Data curation:** Kingori Ndungu.

**Formal analysis:** Kingori Ndungu.

**Investigation:** Kingori Ndungu.

**Methodology:** Kingori Ndungu, Peter Gichangi, Marleen Temmerman.

**Software:** Kingori Ndungu.

**Supervision:** Peter Gichangi, Marleen Temmerman.

**Validation:** Peter Gichangi, Marleen Temmerman.

**Writing – original draft:** Kingori Ndungu.

**Writing – review & editing:** Peter Gichangi, Marleen Temmerman.

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
