## [Decision Letter · Decision Letter 0]

16 Nov 2021

PONE-D-21-07875What determines HIV self-test Acceptability and Uptake within the MSM community in Nairobi and its Environs? A cross sectional studyPLOS ONE

Dear Dr. Ndungu,

Thank you for submitting your manuscript to PLOS ONE. After careful consideration, we feel that it has merit but does not fully meet PLOS ONE’s publication criteria as it currently stands. Therefore, we invite you to submit a revised version of the manuscript that addresses the points raised during the review process.

 Three reviewers have evaluated your submission and have identified a number of concerns that need to be addressed carefully in a revision. Please pay particular attention to responding to the points raised regarding the presentation of the manuscript and the need for additional clarity about your study's limitations.

We look forward to receiving your revised manuscript.

Kind regards,

Jamie Males

Staff Editor

PLOS ONE

Journal Requirements:

2. In the Methods section, please provide additional information regarding the questionnaire development process, including the theories or frameworks which were employed. And finally please include additional details regarding the questionnaire validation.

3. PLOS requires an ORCID iD for the corresponding author in Editorial Manager on papers submitted after December 6th, 2016. Please ensure that you have an ORCID iD and that it is validated in Editorial Manager. To do this, go to ‘Update my Information’ (in the upper left-hand corner of the main menu), and click on the Fetch/Validate link next to the ORCID field. This will take you to the ORCID site and allow you to create a new iD or authenticate a pre-existing iD in Editorial Manager. Please see the following video for instructions on linking an ORCID iD to your Editorial Manager account: https://www.youtube.com/watch?v=_xcclfuvtxQ.

4. Please amend the manuscript submission data (via Edit Submission) to include author, Peter Gichangi, Marleen Temmerman.

Reviewers' comments:

Reviewer's Responses to Questions

**Comments to the Author**

1. Is the manuscript technically sound, and do the data support the conclusions?

Reviewer #1: Partly

Reviewer #2: No

Reviewer #3: No

2. Has the statistical analysis been performed appropriately and rigorously? 

Reviewer #1: No

Reviewer #2: I Don't Know

Reviewer #3: No

3. Have the authors made all data underlying the findings in their manuscript fully available?

Reviewer #1: Yes

Reviewer #2: Yes

Reviewer #3: No

4. Is the manuscript presented in an intelligible fashion and written in standard English?

Reviewer #1: Yes

Reviewer #2: No

Reviewer #3: No

5. Review Comments to the Author

Reviewer #1: I appreciate the opportunity to review and provide feedback on the draft manuscript entitled, “What determines HIV self-test Acceptability and Uptake within the MSM community in Nairobi and its Environs? A cross sectional study” by Ndungu et al. The topical matter of the manuscript is important spanning the burgeoning field of HIV self-testing (HIVST) in a key population of MSM/MSW within an Africa context. I commend the authors for undertaking this research particularly in a setting where the population of interest must exist in non-open manner given the legal statutes. The report has valuable information to put forth in the literature and with appropriate revisions (outlined below) will likely be suitable for publication.

General:

There are numerous typos throughout the draft that should be addressed with careful proofreading. Related there are many instances in the prose where the authors write out a numeric result and then immediately report it as numerals, this is not needed and is redundant. See lines 222, 316, 324, and many more.

The database provided has the participant date of birth listed THIS MUST BE REMOVED AS IT IS PROTECTED HEALTH INFORMATION.

Title:

The title is a bit definitive, suggesting that the report will answer the question posed. A preferable phrasing might be “Evaluation of factors associated with HIV self-test Acceptability and Uptake within the MSM community in Nairobi, Kenya: A cross sectional study”

Abstract: The last sentence of the background is a fragment and unclear. The objective seems to be to identify both facilitating and barrier factors for HIVST aspects the way written it only states “facilitating”. There is no description of the analysis here there needs to be at least some. The conclusion that “A significant number of MSM community in Kenya are willing to use HIVST and are likely to seek for care within 30 days and this is a good indicator of linkage” is a bold statement given the data and this should be more tempered given the sample studied.

Introduction: There are a few typos that need to be addressed but overall is well constructed and appropriate for the work.

Methods: Need more detail in all sections. For the Study population it could not have been “all adult men aged 16-60” years as the author’s state. The last two sentences in this section are introduction information and do not belong in the methods section. The sampling technique is not clear and needs more details on when the sampling occurred, over what time period and with what frequency for each of the sites. As well the snowball technique is not a standard approach for a cross-sectional design and is more often used in qualitative sampling so it would be good to have more rational and information on how this was achieved. How was the questionaries administered was it in English or Kiswahili or a different language? The analysis choice to use POR would benefit from some reasoning as to why. It is not wrong to do so mathematically but not overly common so some more information for the reader would be preferable. The number of inferential comparisons preformed is immense (~20) and as such the authors should adjust down their p-value used to make assertions around assumptions and “significant” associations. A simple Bonferroni correction would be acceptable but there are more advanced methods that could be chosen. The reporting of the POR

Results: The prose section of the results is long and can be cut down, the authors should focus in on highlighting only the most key findings in this section. To achieve this focusing in on the inferential analysis with the proportional differences between comparison groups and the PORs would be helpful. As well for all numeric values there is no need to provide more than three significant digits (save for p values <0.001). When describing a POR that is a finding showing lower likelihood of HIVST use it would be easier to understand if the result was framed as a value <1 (see lines 332,333 for example).

Discussion: The discussion raises good points throughout however the authors should aim to be more tempered in their interpretation of the results given the overall limitations. I am not sure if the sub-headings are needed but that is up to the journal. The limitations section should be expanded on there is most certainly selection bias given the sampling approach (snowball) and although this is mentioned more elaboration on this point is needed. Additionally consideration of responder bias and the impact of missing data should be discussed.

Tables and Figures:

Given the design a figure 1 flow diagram to understand those that were screened and did not consent/participate as well as the 22 who were lost to follow up and did not complete data collection.

Table 3 is entirely too large to be digestible by the reader. Recommend breaking up the inferential portions of the analyses and reporting them as separate tables.

Reviewer #2: The article had important problems regarding language.

The use of numbers in parenthesis should be revised. For example, authors said: "The majority of interviewees were Kenyans at (92.6%), followed....".

Numbers of the tables should be revised (Table 1 does not exist).

I do not understand how 6 participants that "used HIV test kit before", answer "no" when asked about "Ever tested for HIV". How did they used the self-test if they were never tested?.

In the same way, 136 participants that "used HIV test kit before", answer "no" when asked about "Ever heard of HIV self test". How did they used the self-test if they never heard about it? It does not make sense.

Reviewer #3: The authors aimed to determine facilitating factors for HIVST acceptability and uptake among MSM community. This area is of great interest and important, however, the manuscript will need significant improvement

Major comments

On abstract

Introduction

• Integrate objective within the introduction section and remove sub heading objective

Methods

• Add the study outcome of interest and how it was defined or measured, and statistical methods used to analyze the data, this will help simplify follow up of the results

Results

• The objective was to determine facilitating factors for HIVST acceptability and uptake among MSM community, however, I don’t see results that support this objective. It looks like you did a simple descriptive analysis. Is it possible to do a regression model on data collected? You present hindrance to testing more than facilitators

Conclusion

• Conclusion is not backed by results and not aligned with the main objective and title of the study of the study. Please, clarify

Background

• The opening statement needs figures to back that urgent of “There has been significant progress in HIV prevention efforts across Africa, however men who have sex with men (MSM) continue to bear a disproportionately heavy burden of HIV infection compared to the general adult population”.

• Reference two is so old, please, cite the most recent data and citation

• The back ground is too long, and not clearly written in line with objective and title of study. I suggest, they write in funnel shape- start with MSM global HIV risk and testing rates, narrowing to Kenya, then to HIV self-testing acceptability and uptake, finally state the clear problem they intent to address/study and end with what they did to address this problem

Methods

• Start with study design, then study area etc. I have seen study design is under sampling techniques which makes following up of the manuscript difficult.

• Under data analysis, I think it can be improved better by using a suitable regression model depending on how they will define their outcome of interest

• Results too are too long, I suggest the author aligns the results to study objectives and topic and concentrate on that. For now, the authors present a lot of data which is good, but should be aligned to the topic/objectives. The results are broad ranging from HIVST Acceptability, Uptake, Facilitators, willingness and Barriers

Discussion

• The entire opening paragraph is not necessary. The first paragraph should summarize the main findings of your study and then you continue to compare it with other published studies.

• Again, the discussion is very long and discusses different themes that are not aligned with the topic and objective of the study. Too much data presented

• Conclusions needs to be aligned with topic and main objectives

6. PLOS authors have the option to publish the peer review history of their article (what does this mean?). If published, this will include your full peer review and any attached files.

Reviewer #1: **Yes: **Adam R. Aluisio

Reviewer #2: No

Reviewer #3: No

---

## [Author Response · Author response to Decision Letter 0]

4 Jun 2022

Reviewers & Authors Comments.

PONE-D-21-07875

What determines HIV self-test Acceptability and Uptake within the MSM community in Nairobi and its Environs? A cross sectional study

PLOS ONE

Dear Dr. Ndungu,

Thank you for submitting your manuscript to PLOS ONE. After careful consideration, we feel that it has merit but does not fully meet PLOS ONE’s publication criteria as it currently stands. Therefore, we invite you to submit a revised version of the manuscript that addresses the points raised during the review process.

Three reviewers have evaluated your submission and have identified a number of concerns that need to be addressed carefully in a revision. Please pay particular attention to responding to the points raised regarding the presentation of the manuscript and the need for additional clarity about your study's limitations.

Please submit your revised manuscript. If you will need more time than this to complete your revisions, please reply to this message or contact the journal office at plosone@plos.org. Reviewer #1

2. In the Methods section, please provide additional information regarding the questionnaire development process, including the theories or frameworks which were employed. And finally, please include additional details regarding the questionnaire validation.

Response

The questionnaire was developed through a consultative, participatory approach by consulting experts in the subject matter. The questionnaire development process was also informed by previous research done on the thematic area. References have been provided. The questionnaire was validated through a pre-test. The researchers also employed a Theoretical Framework combining AIDS Risk Reduction Model (ARRM) and Modified Social Ecological Model showing the interconnectedness of the different factors. (See attached the Theoretical framework).

3. PLOS requires an ORCID iD for the corresponding author in Editorial Manager on papers submitted after December 6th, 2016. Please ensure that you have an ORCID iD and that it is validated in Editorial Manager. To do this, go to ‘Update my Information’ (in the upper left-hand corner of the main menu), and click on the Fetch/Validate link next to the ORCID field. This will take you to the ORCID site and allow you to create a new iD or authenticate a pre-existing iD in Editorial Manager. Please see the following video for instructions on linking an ORCID iD to your Editorial Manager account: https://www.youtube.com/watch?v=_xcclfuvtxQ.

4. Please amend the manuscript submission data (via Edit Submission) to include author, Peter Gichangi, Marleen Temmerman.

 Technical Comments:

Reviewer #2: I appreciate the opportunity to review and provide feedback on the draft manuscript entitled, “What determines HIV self-test Acceptability and Uptake within the MSM community in Nairobi and its Environs? A cross sectional study” by Ndungu et al. The topical matter of the manuscript is important spanning the burgeoning field of HIV self-testing (HIVST) in a key population of MSM/MSW within an Africa context. I commend the authors for undertaking this research particularly in a setting where the population of interest must exist in non-open manner given the legal statutes. The report has valuable information to put forth in the literature and with appropriate revisions (outlined below) will likely be suitable for publication.

Reviewers Comments Response

Reviewer # 2 

General

There are numerous typos throughout the draft that should be addressed with careful proofreading. Related there are many instances in the prose where the authors write out a numeric result and then immediately report it as numerals, this is not needed and is redundant. See lines 222, 316, 324, and many more.

The database provided has the participant date of birth listed THIS MUST BE REMOVED AS IT IS PROTECTED HEALTH INFORMATION. The researchers have taken time to carefully go through the paper and addressed the typos. The redundancy through repetitions of numerals and reporting on the same has been addressed. The date on birth on the database has been removed

Title:

The title is a bit definitive, suggesting that the report will answer the question posed. A preferable phrasing might be “Evaluation of factors associated with HIV self-test Acceptability and Uptake within the MSM community in Nairobi, Kenya: A cross sectional study The researchers have deliberated on the papers title and have decided to adopt the below;

Evaluation of factors associated with HIV self-test Acceptability and Uptake within the MSM community in Nairobi and its environs, Kenya: A cross sectional study

Abstract:

The last sentence of the background is a fragment and unclear. The objective seems to be to identify both facilitating and barrier factors for HIVST aspects the way written it only states “facilitating”. There is no description of the analysis here there needs to be at least some. The conclusion that “A significant number of MSM community in Kenya are willing to use HIVST and are likely to seek for care within 30 days and this is a good indicator of linkage” is a bold statement given the data and this should be more tempered given the sample studied. The authors have noted the comments, and reorganized the background as well as tempered the conclusion

Introduction:

There are a few typos that need to be addressed but overall is well constructed and appropriate for the work. The typos have been addressed

Methods:

Need more detail in all sections. For the Study population it could not have been “all adult men aged 16-60” years as the author’s state. The last two sentences in this section are introduction information and do not belong in the methods section. The sampling technique is not clear and needs more details on when the sampling occurred, over what time period and with what frequency for each of the sites. As well the snowball technique is not a standard approach for a cross-sectional design and is more often used in qualitative sampling so it would be good to have more rational and information on how this was achieved. How was the questionaries’ administered was it in English or Kiswahili or a different language? The analysis choice to use POR would benefit from some reasoning as to why. It is not wrong to do so mathematically but not overly common so some more information for the reader would be preferable. The number of inferential comparisons preformed is immense (~20) and as such the authors should adjust down their p-value used to make assertions around assumptions and “significant” associations. A simple Bonferroni correction would be acceptable but there are more advanced methods that could be chosen. The reporting of the POR 

The methods have been carefully looked at and better presented. The study population comprised MSM aged 18 and above to allow for individual informed consent. The reason for excluding minors from the study has also been well presented. The sampling techniques has been redone. An attempt has also been made to explain reasons for choice of sampling methods.

The researchers distributed English paper based self-administered structured questionnaires between July 2018-June 2019 after obtaining oral informed consent.

The researcher’s used POR because we wanted to know the odds on the different risk and the data collection was cross-sectional, hence we would not use direct odds ratio. The P-Value has been adjusted in the revised paper.

Results: The prose section of the results is long and can be cut down, the authors should focus in on highlighting only the most key findings in this section. To achieve this focusing in on the inferential analysis with the proportional differences between comparison groups and the PORs would be helpful. As well for all numeric values there is no need to provide more than three significant digits (save for p values <0.001). When describing a POR that is a finding showing lower likelihood of HIVST use it would be easier to understand if the result was framed as a value <1 (see lines 332,333 for example). The results section has been trimmed down and only the key variables have been captured.

Inferential analysis has been done and better expounded on the revised paper

Discussion: The discussion raises good points throughout however the authors should aim to be more tempered in their interpretation of the results given the overall limitations. I am not sure if the sub-headings are needed but that is up to the journal. The limitations section should be expanded on there is most certainly selection bias given the sampling approach (snowball) and although this is mentioned more elaboration on this point is needed. Additionally, consideration of responder bias and the impact of missing data should be discussed. The discussion section has been re-organized and its now more precise. The subheadings have also been dropped. The limitation section has been better expounded and how the sampling bias was addressed.

Explanations on missing data have been provided

Tables and Figures:

Given the design a figure 1 flow diagram to understand those that were screened and did not consent/participate as well as the 22 who were lost to follow up and did not complete data collection.

Table 3 is entirely too large to be digestible by the reader. Recommend breaking up the inferential portions of the analyses and reporting them as separate tables. 

This section has been better presented and requested information provided

All the tables including table 3 have been re-analyzed

Reviewer # 3: 

The article had important problems regarding language.

The use of numbers in parenthesis should be revised. For example, authors said: "The majority of interviewees were Kenyans at (92.6%), followed....".

Numbers of the tables should be revised (Table 1 does not exist).

I do not understand how 6 participants that "used HIV test kit before", answer "no" when asked about "Ever tested for HIV". How did they used the self-test if they were never tested?.

In the same way, 136 participants that "used HIV test kit before", answer "no" when asked about "Ever heard of HIV self-test". How did they used the self-test if they never heard about it? It does not make sense The language has been adjusted throughout the paper, and presentation. The tables have been properly numbered, and the figures re-checked

Reviewer # 4: 

The authors aimed to determine facilitating factors for HIVST acceptability and uptake among MSM community. This area is of great interest and important, however, the manuscript will need significant improvement 

Introduction

• Integrate objective within the introduction section and remove sub heading objective The authors have adjusted the introduction accordingly

Methods

• Add the study outcome of interest and how it was defined or measured, and statistical methods used to analyze the data, this will help simplify follow up of the results The variables of interest have been enumerated and statistical methods employed better expounded.

Results

• The objective was to determine facilitating factors for HIVST acceptability and uptake among MSM community, however, I don’t see results that support this objective. It looks like you did a simple descriptive analysis. Is it possible to do a regression model on data collected? You present hindrance to testing more than facilitators The findings have been reorganized to reflect the objectives of the study

Conclusion

• Conclusion is not backed by results and not aligned with the main objective and title of the study of the study. Please, clarify The authors have re-written the conclusions

Background

The opening statement needs figures to back that urgent of “There has been significant progress in HIV prevention efforts across Africa, however men who have sex with men (MSM) continue to bear a disproportionately heavy burden of HIV infection compared to the general adult population”.

• Reference two is so old, please, cite the most recent data and citation

• The back ground is too long, and not clearly written in line with objective and title of study. I suggest, they write in funnel shape- start with MSM global HIV risk and testing rates, narrowing to Kenya, then to HIV self-testing acceptability and uptake, finally state the clear problem they intent to address/study and end with what they did to address this problem The authors have included additional citations to support the opening statement. Additionally, explanations about the rates and infection differentials have been expounded in the subsequent text.

The background has been rewritten accordingly

Methods

• Start with study design, then study area etc. I have seen study design is under sampling techniques which makes following up of the manuscript difficult.

• Under data analysis, I think it can be improved better by using a suitable regression model depending on how they will define their outcome of interest

• Results too are too long, I suggest the author aligns the results to study objectives and topic and concentrate on that. For now, the authors present a lot of data which is good, but should be aligned to the topic/objectives. The results are broad ranging from HIVST Acceptability, Uptake, Facilitators, willingness and Barriers The authors have adjusted the methods section as per the reviewer’s suggestions. The results have also been adjusted to reflect the study objectives.

Discussion

• The entire opening paragraph is not necessary. The first paragraph should summarize the main findings of your study and then you continue to compare it with other published studies.

• Again, the discussion is very long and discusses different themes that are not aligned with the topic and objective of the study. Too much data presented

• Conclusions needs to be aligned with topic and main objectives The authors have adjusted the discussion section and attempted to align it to the objectives of the study. Data has also been reduced and conclusion better aligned

---

## [Editor Report · Decision Letter 1]

14 Jun 2022

PONE-D-21-07875R1Reviewers & Authors Comments.

Evaluation of factors associated with HIV self-testing Acceptability and Uptake within the MSM community in Nairobi, Kenya: A cross sectional study.PLOS ONE

Dear Dr. Ndungu,

Thank you for submitting your manuscript to PLOS ONE. After careful consideration, we feel that it has merit but does not fully meet PLOS ONE’s publication criteria as it currently stands. Therefore, we invite you to submit a revised version of the manuscript that addresses the points raised during the review process.

We look forward to receiving your revised manuscript.

Kind regards,

Adam R Aluisio, M.D MSc., DTM&H

Guest Editor

PLOS ONE

Additional Editor Comments (if provided):

I served as an initial reviewer on this submission and now as a guest editor. Good progress on this work but there was inadequate response to some of the reviewers' comments. Specifically no information on bias or missing data although the response letter states this was done. As the primary outcome is based on self-report of past HIVST use this needs to be acknowledged at least in the limitations. Also the main outcome metric of HIV self-testing needs to be better described in the methods. It seems that is based on participant report solely, which is fine but needs to be more clearly stated and how that also could suffer from bias. The updated draft with the regression model is difficult to follow on how it was constructed and run. Associated with that the reporting of the model is confusing as to what the main findings are and the tables have too much information on statical outputs (there is no need to report coefficients and the aPOR or the SE and the CIs). These aspects need to be corrected for publication.
---

## [Author Response · Author response to Decision Letter 1]

30 Jul 2022

Reviewer Comments & Response

Reviewers Comments Response

Reviewer #1 

Specifically, no information on bias or missing data although the response letter states this was done. 

 This comment is well received.

Statement on management of missing data on the questionnaires has been included in the abstract (line 41-43) and the methods (line 171-177).

As the primary outcome is based on self-report of past HIVST use this needs to be acknowledged at least in the limitations.

 The study is based on respondent self-reports on HIVST use, and as such the veracity of such reports could not be easily determined within the design of this study. The above statement has been included in the discussion (Line 399-401)

The main outcome metric of HIV self-testing needs to be better described in the methods.

It seems that is based on participant report solely, which is fine but needs to be more clearly stated and how that also could suffer from bias

 The operational definition of HIV self-testing has been in-cooperated in the methods (Line 192-197)

The updated draft with the regression model is difficult to follow on how it was constructed and run. Associated with that the reporting of the model is confusing as to what the main findings are and the tables have too much information on statical outputs (there is no need to report coefficients and the aPOR or the SE and the CIs). These aspects need to be corrected for publication.

 The multivariate analysis has been explained how it was constructed and organized and was based on measure of association (Chi-Square) and risk ratio (POR) captured in the multivariate analysis (Line 284-285).

The table has also been reduced and only key outputs have been retained. Columns on S.E, Wald have been deleted

---

## [Editor Report · Decision Letter 2]

12 Aug 2022

PONE-D-21-07875R2Evaluation of factors associated with HIV self-testing Acceptability and Uptake within the MSM community in Nairobi, Kenya: A cross sectional study.PLOS ONE

Dear Dr. Ndungu,

Thank you for submitting your manuscript to PLOS ONE. After careful consideration, we feel that it has merit but does not fully meet PLOS ONE’s publication criteria as it currently stands. Therefore, we invite you to submit a revised version of the manuscript that addresses the points raised during the review process.

We look forward to receiving your revised manuscript.

Kind regards,

Adam R Aluisio, M.D MSc., DTM&H

Guest Editor

PLOS ONE

Journal Requirements:

Additional Editor Comments (if provided):

Good progress by the author team. This is an improved draft.

Two required points to revise.

1.) The authors state the missing data were MCAR to justify the listwise deletion. Please provide the information on how this was assessed and add that to the methods and/or results as appropriate. If a graphical method was utilized this can be provided as a supplement for the readers.

2.) The tables are more understandable in this format. For the aPOR table please indicate (perhaps footnotes) what the reference categories are. All tables and figures should be interpretable by themselves.

Reviewers' comments:

N/A

---

## [Author Response · Author response to Decision Letter 2]

21 Dec 2022

PONE-D-21-07875R2

Evaluation of factors associated with HIV self-testing Acceptability and Uptake within the MSM community in Nairobi, Kenya: A cross sectional study.

Reviewers & Authors Comments.

Good progress by the author team. This is an improved draft.

Reviewers Comments Response

Journal Requirements:

The references have been carefully checked to ensure they are complete and non-has been retracted. 

Reviewer #1

The authors state the missing data were MCAR to justify the listwise deletion. Please provide the information on how this was assessed and add that to the methods and/or results as appropriate. If a graphical method was utilized this can be provided as a supplement for the readers.

 Graphical method was utilized to inform the adoption of listwise deletion. Graphs have been attached in the appendix 

The tables are more understandable in this format. For the a POR table please indicate (perhaps footnotes) what the reference categories are. All tables and figures should be interpretable by themselves.

 The comment is well received. All the tables have been reformatted to be self -explanatory We have Introduced borders and the Ref. to distinguish the reference variables.

---

## [Editor Report · Decision Letter 3]

4 Jan 2023

Evaluation of factors associated with HIV self-testing Acceptability and Uptake among the MSM community in Nairobi, Kenya: A cross sectional study.

PONE-D-21-07875R3

Dear Dr. Ndungu,

We’re pleased to inform you that your manuscript has been judged scientifically suitable for publication and will be formally accepted for publication once it meets all outstanding technical requirements.

Kind regards,

Adam R Aluisio, M.D MSc., DTM&H

Guest Editor

PLOS ONE

Additional Editor Comments (optional):

The revisions have been completed with no additional major concerns. I thank the authors for their hard work and contribution to the field.
---

## [Editor Report · Acceptance letter]

27 Feb 2023

PONE-D-21-07875R3 

Evaluation of factors associated with HIV self-testing Acceptability and Uptake among the MSM community in Nairobi, Kenya: A cross sectional study. 

Dear Dr. Ndungu:

I'm pleased to inform you that your manuscript has been deemed suitable for publication in PLOS ONE. Congratulations! Your manuscript is now with our production department. 

Kind regards, 

on behalf of

Dr. Adam R Aluisio 

Guest Editor

PLOS ONE